∂ | **Open Peer Review** | Environmental Microbiology | Observation

# HighALPS: ultra-high-throughput marker-gene amplicon library preparation and sequencing on the Illumina NextSeq and NovaSeq Platforms

Lena Flörl,[1] Paula Momo Cabrera,[1] Maria Domenica Moccia,[2] Serafina Plüss,[1] Nicholas A. Bokulich[1]

**ABSTRACT** Microbiome research using amplicon sequencing of microbial marker genes has surged over the past decade, propelled by protocols for highly multiplexed sequencing with barcoded primer constructs. Newer Illumina platforms like the NovaSeq and NextSeq series significantly outperform older sequencers in terms of reads, output, and runtime. However, these platforms are more prone to index-hopping, which limits the application of protocols designed for older platforms such as the Earth Microbiome Project protocols; hence, there is a need to adapt these established protocols. Here, we present an ultra-<u>h</u>igh-throughput <u>a</u>mplicon <u>l</u>ibrary <u>p</u>reparation and <u>s</u>equencing protocol (HighALPS) incorporating the capabilities of these newer sequencing platforms, designed for both 16S rRNA gene and fungal internal transcribed spacer domain sequencing. Our results demonstrate good run performance across different sequencing platforms and flow cells, with successful sequencing of mock communities, validating the protocol's effectiveness. The HighALPS library preparation method offers a robust, cost-effective, and ultra-high-throughput solution for microbiome research, compatible with the latest sequencing technologies. This protocol allows multiplexing thousands of samples in a single run at a read depth of tens of millions of sequences per sample.

**IMPORTANCE** Marker gene amplicon sequencing on Illumina devices remains the most commonly used technology to profile microbial communities. Yet, most library preparation protocols are not adapted to harness the capabilities and deal with the caveats of the latest Illumina sequencing platforms, which highly outperform older platforms in terms of speed, quality, and output. Here, we present an ultra-high-throughput, cost-effective, and robust library preparation protocol (HighALPS) optimized to fully leverage the capabilities of the latest Illumina sequencing platforms. The combinatorial unique dual index strategy effectively combats miss-assignment of reads due to index-hopping, which is more prevalent in newer platforms. The HighALPS protocol incorporates technological (e.g., novel sequencing chemistry and lab automation platforms) as well as bioinformatics advances (e.g., denoising algorithms which make triplicate amplifications unnecessary) of the last few years to optimize and streamline library preparation for bacterial and fungal communities.

**KEYWORDS** amplicon library preparation, microbiome, protocol, high-throughput sequencing, 16S rRNA gene sequencing, fungal ITS sequencing, multiplexed sequencing protocols, index-hopping, NovaSeq platform, NextSeq platform

Interest in microbiome research has surged in the past decade, fueled by increasing recognition of the pivotal role that microbiomes play in global ecosystems, including human health. The most common technology used to study microbial communities is marker-gene amplicon sequencing, such as sequencing of the 16S rRNA genes. This

Address correspondence to Nicholas A. Bokulich, Nicholas.Bokulich@hest.ethz.ch.

The authors declare no conflict of interest.

See the funding table on p. 5.

is due to the relatively low cost and high throughput of this approach, as well as the availability of mature software pipelines that facilitate relatively rapid analysis of sequencing data (1).

The popularity of marker-gene sequencing first surged in the early 2010s with the publication of protocols for high-throughput 16S rRNA sequencing performed using Roche 454 Pyrosequencing (2) and Illumina HiSeq and MiSeq platforms (3, 4). A commonly used standard library preparation workflow uses proprietary Illumina Nextera kits and relies on tagmentation of DNA (e.g., marker-gene amplicons) with a set of Nextera unique dual indices (each 8 nt) embedded in the adapter sequence (see Fig. 1), allowing up to 384 samples to be multiplexed and sequenced in a single run. Conversely, the Earth Microbiome Project (EMP) protocol uses unique indices (12 nt-long Golay error-correcting barcodes) that are embedded in the primer constructs and incorporated into the amplicons during PCR amplification (see Fig. 1). This strategy increases the throughput by allowing for a substantially larger number of samples to be pooled into a single run. Additionally, time and labor are reduced, as only a single PCR is necessary. After its original release in 2012, the EMP protocol was slightly updated, and a second version was published in 2023. However, the protocol is still as originally designed, primarily applicable for Illumina MiSeq and HiSeq, which were launched in 2010 and 2011. These protocols are not directly transferable to newer Illumina platforms, such as the NovaSeq and NextSeq series, as the patterned flow cells used by these devices have a higher risk of index hopping (5), which therefore benefit from a unique dual index (UDI) strategy to minimize read miss-assignment. Yet, these newer platforms massively outperform the older sequencers in regard to maximal read output and run time (see Table 1), leading to dramatically reduced costs per gigabase.

Here, we introduce a new ultra-High-throughput Amplicon Library Preparation and Sequencing method (HighALPS) to profile microbial communities, based on the principles of the EMP protocol, and optimized for cost-effectiveness and compatibility with newer Illumina DNA sequencing platforms. HighALPS is robust against the effects of index hopping as the combinatorial UDI strategy enables the removal of unexpected index combinations generated by index hopping. While the protocol does not reduce the occurrence of index hopping itself, it ensures that hopped reads can be identified and excluded during demultiplexing. This form of sequencing error is more common in Illumina patterned flow cells and typically ranges between 0.1% and 2% of reads affected (5). Such errors can significantly compromise downstream analyses and, depending on the study design, may be detrimental. In single-index libraries such as the original EMP protocol, this would lead to a proportion of incorrectly assigned reads in demultiplexing. Using a combinatorial UDI approach significantly reduces the likelihood of sample misassignments, with the specific rate depending on the number of barcodes and combinations thereof. For instance, with the proposed combination (see Fig. S1), the maximum theoretical misassignment rate is 0.04%. Notably, while it is highly recommended to rely on a UDI strategy when using patterned flow cells, this combinatorial UDI protocol is also compatible with other Illumina platforms, as most flow cells support the addition of custom sequencing primers. For example, a HighALPS library can be run on a MiniSeq for shallow sequencing prior to pooling, or a MiSeq when processing a smaller number of samples. Further, the combinatorial UDI strategy is particularly economical, as it requires a lower number of individually barcoded primers, which due to the purification requirements and length are more costly. For example, the combination of only 96 unique forward and reverse primers (see Fig. S1) can result in 1,152 unique indices. The cost for these combinatorial UDI primers is approximately ~0.34 USD (based on current rates) per 96-plate, which is less than 1% of the cost in comparison to using the proprietary Illumina Nextera XT Index Kit v2 Set for ~409.55 USD per 96-plate (see File S1.3). Additionally, when comparing the cost per sequencing run using appropriate flow cells (500–600 cycles) across older and newer Illumina platforms, the cost of a NovaSeq 6000 or NextSeq 1000/2000 run ranges from 7.7% to 20.3% of the price of a MiSeq run per million reads obtained, based on current rates. Similarly, the cost per gigabyte of

**TABLE 1** A comparison of Illumina sequencers based on their release years shows differences in output, run time, read length, and the underlying technologies

| Platform | Launched | Max reads per run | Max output per flow cell | Run time | Max read length | Chemistry | Flow cell |
|---|---|---|---|---|---|---|---|
| NovaSeq X Series | 2022 | 52 B[a] | 8 TB | ~17–48 h | 2 × 150 bp | XLEAP-SBS | Ultra high-density patterned flow cell |
| NextSeq 1000 and 2000 | 2020 | 1.8 B | 540 GB | ~8–44 h | 2 × 300 bp | XLEAP-SBS | Patterned flow cell |
| NovaSeq 6000 | 2017 | 20 B[a] | 3 TB | ~13–44 h | 2 × 250 bp | SBS | Patterned flow cell |
| MiSeq | 2011 | 25 M | 15 GB | ~4–56 h | 2 × 300 bp | SBS | Non-patterned flow cells |
| HiSeq | 2010 | 3 B[a] | 600 GB | ~11 days | 1 × 100 bp | SBS | Non-patterned flow cells |

[a]Data output for dual flow cells (6).

output data for the NovaSeq 6000 and NextSeq 1000/2000 ranges from only 4.6% to 20.3% of that of a MiSeq run (see File S3.1). Further, as these library preparation primers are designed for each marker gene separately, amplicons of different marker genes can be multiplexed in a single run (e.g., for simultaneous sequencing of 16S rRNA gene and internal transcribed spacer [ITS] amplicons). This additional multiplexing does not only reduce costs through enabling combined sequencing runs but increases sequence diversity within the run, which allows for the reduction of PhiX and can contribute to overall higher read quality.

Detailed methodology for the HighALPS protocol development, considerations, and validation thereof can be found in the File S1. Briefly, we designed library preparation primer constructs to profile bacterial communities incorporating the commonly used 515F (5′-GTGYCAGCMGCCGCGGTAA-3′) (8) and 806R primer pair (5′-GGACTACNVGGGTWTCTAAT-3′) (9). These target the hypervariable V4 region of the 16S rRNA gene and are the same primers used in the original EMP protocol (4). For fungal ITS sequencing, we designed constructs targeting the ITS1 domain with the following

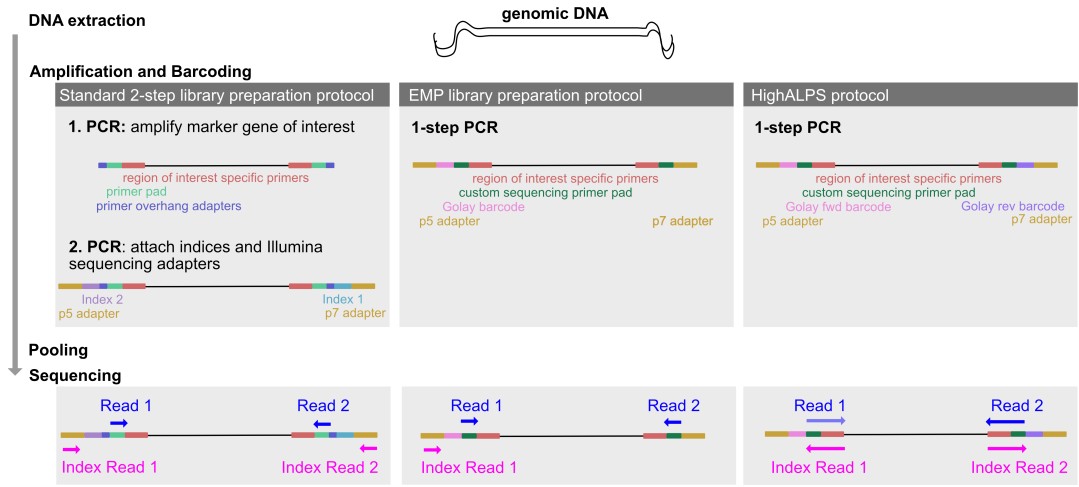

**FIG 1** Comparison of different library preparation strategies. The genomic DNA to be amplified is depicted as the black strand and the colored blocks indicate the primer constructs. (i) Amplification and barcoding. For the standard two-step NGS amplicon library preparation protocol using an Illumina Nextera kit, in the first PCR the marker gene of interest (GOI) is amplified with a primer for the region of interest (red), linked to the primer pad (green), where the sequencing primers ultimately bind, as well as an overhang adapter (blue) for the second PCR. In the subsequent second PCR, which typically entails only 8–10 cycles, two barcodes for unique dual indexing (purple, light blue) as well as the flow cell adapters (gold) are attached. These p5 and p7 adapters attach the nucleotide strand to the flow cell and are universal between all Illumina instruments (7). In comparison, the EMP protocol requires only a single PCR reaction as the primer constructs already contain the adapters, a unique barcode in the forward primer, a custom primer pad, as well as the GOI-specific primers. Similarly, the HighALPS protocol requires only a single PCR step; however, the forward primer as well as the reverse primer carry a unique barcode that enables combinatorial unique dual indexing. (ii) Sequencing. In older Illumina platforms, index reads were generated from primers anchored to the adapters. In contrast, dual-indexed sequencing runs on NovaSeq and NextSeq platforms employ a Reverse Complement Workflow. The index primers, therefore, are the reverse complements of the read primers, that is, index primer 1 being the reverse complement of read primer 2 and vice versa.

**TABLE 2**  Key run quality parameters for the sequencing of libraries created with HighALPS[a]

| Platform | Flowcell | Yield (GBases) | % ≥ Q30 bases | % clusters occupied | % clusters passing filter |
|---|---|---|---|---|---|
| NextSeq 2000 | P1, 600 cycles | 81.98 | 90.22 | 91.23 | 81.55 |
| NovaSeq 6000 | SP, 500 cycles | 490.72 | 85.59 | 94.78 | 73.61 |

[a]Run performance is considered good when at least 80% of reads have a quality score of 30 or higher (% ≥ Q30 bases) (12) which both runs easily surpassed. Both runs also show a fairly high amount of percentage of clusters occupied and clusters passing the filter. A low percentage of clusters passing filter could be due to a low library quality and or over clustering, and values above 60–65% are typically considered successful runs (13).

primers: BITS (5′-ACCTGCGGARGGATCA-3′) and B58S3 (5′-GAGATCCRTTGYTRAAAGTT-3′), which demonstrate high coverage of most fungal groups (10). However, in theory, any other marker-gene primers suitable for Illumina short-read sequencing could be substituted to target, including other 16S rRNA gene domains, fungal ITS2, or other targets (e.g., CO1 for diet metabarcoding). For these constructs, we assessed primer interactions and dimerization potential. Library preparation is performed in a single-step PCR, significantly reducing both time and reagent costs compared to a standard two-step library preparation protocol, which requires two PCRs, and the original EMP protocol, where samples were amplified in triplicates. Given the increased robustness of modern bioinformatic tools in detecting chimeras and jackpot effects, triplicate amplification is no longer necessary (11). Ultimately, this library is sequenced with custom sequencing primers, optimized for the NovaSeq 6000 or NextSeq 2000 platform (see Fig. 1).

We tested the novel HighALPS library preparation method across different sequencing platforms and flow cells, which resulted in good run performances. Specifically, we applied the described method on the NextSeq 2000 and NovaSeq 6000 sequencing platforms using different flow cells (P1 and SP) (see Table 2). Further, we validated HighALPS with mock communities of various bacteria and yeast as described in File S1. Both marker genes for profiling bacterial and fungal communities were sequenced within the same run (see run performances in Table 2) and the retrieved bacterial and fungal genera robustly match the theoretical composition of the mock community (see Fig. 2).

In summary, we show that the new HighALPS ultra-high-throughput library preparation and sequencing method can be robustly used across both Illumina Novaseq and Nextseq Series platforms and effectively profiles microbial communities. In contrast to using individual unique barcodes, a combinatorial UDI strategy is adopted to counteract sample misassignment due to index-hopping, as well as to increase

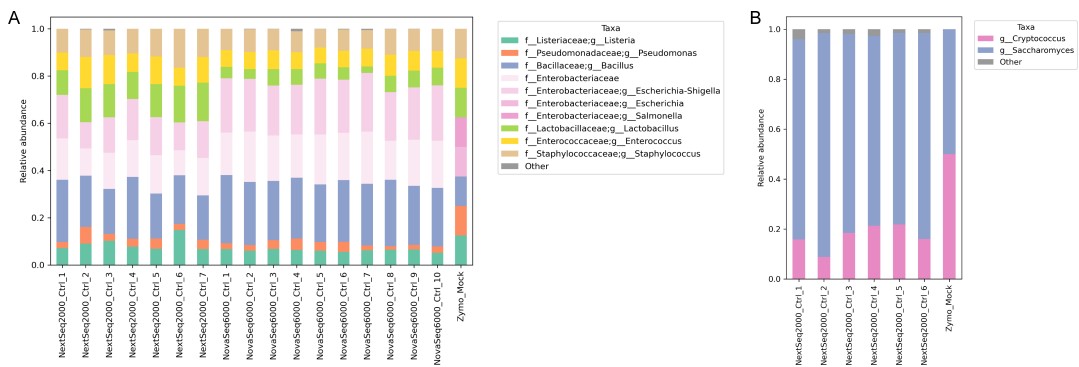

**FIG 2**  Taxon barplots of the microbial features retrieved in comparison to the theoretical composition based on genomic DNA concentration of the mock community (Zymo_Mock, very right bar, respectively) show a consistent pattern of bacterial (A) and fungal (B) genera across both runs. Notably, we observe the widely reported limitation in taxonomically resolving the Enterobacteriaceae family as these species share highly similar 16S rRNA genes. Differences between the theoretical composition and retrieved composition are likely due to differences in DNA extraction efficiencies, copy number variations, and PCR amplification bias that is independent of the sequencing platform used. Particularly for fungal ITS reads, the disparity between expected and observed relative frequencies most likely reflects the wide variation in ITS copy numbers in different fungal species, rather than sequencing-based effects.

cost-effectiveness. Further, HighALPS enables multiplexing of multiple marker genes, which additionally reduces costs and supports higher sequencing data quality due to increased diversity within the run. We provide a detailed step-by-step protocol with automation options and practical tips, as well as custom primer constructs for library preparation and sequencing. In the future, ultra-high-throughput library preparation protocols, such as the presented HighALPS, and decreasing sequencing costs will make microbiome research more accessible to a broader research community.

## ACKNOWLEDGMENTS

We thank the Genomic Diversity Center (GDC) of ETH Zürich as well as the Functional Genomics Center Zürich (FGCZ) for their support.

This work was financially supported by the Swiss National Science Foundation (Grant number: 310030_204275) (to N.A.B.) and the Swiss Government Excellence PhD Scholarship (to L.F.).

## AUTHOR AFFILIATIONS

[1]Department of Health Sciences and Technology, ETH Zurich, Zurich, Switzerland
[2]Functional Genomics Center Zürich (FGCZ), ETH Zurich and University of Zurich, Zurich, Switzerland

## AUTHOR ORCIDs

Lena Flörl http://orcid.org/0000-0003-4458-4071
Nicholas A. Bokulich http://orcid.org/0000-0002-1784-8935

## FUNDING

| Funder | Grant(s) | Author(s) |
| --- | --- | --- |
| Schweizerischer Nationalfonds zur Förderung der Wissenschaftlichen Forschung | 310030_204275 | Nicholas A. Bokulich |

## AUTHOR CONTRIBUTIONS

Lena Flörl, Conceptualization, Data curation, Methodology, Validation, Writing – original draft, Writing – review and editing | Paula Momo Cabrera, Conceptualization, Methodology, Validation, Writing – review and editing | Maria Domenica Moccia, Conceptualization, Methodology, Writing – review and editing | Serafina Plüss, Methodology, Writing – review and editing | Nicholas A. Bokulich, Conceptualization, Funding acquisition, Methodology, Project administration, Supervision, Writing – review and editing

## ADDITIONAL FILES

The following material is available online.

### Supplemental Material

**Supplemental Material (mSystems00023-26-S0001.pdf).** Supplemental figures, tables, and methodology.

### Open Peer Review

**PEER REVIEW HISTORY (review-history.pdf).** An accounting of the reviewer comments and feedback.

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
