## [Reviewer comments · mSystems]

HighALPS: Ultra-High-Throughput Marker-Gene Amplicon Library Preparation and Sequencing on the Illumina NextSeq and NovaSeq Platforms

Lena Flörl, Paula Cabrera, Maria Moccia, Serafina Plüss, and Nicholas Bokulich

Corresponding Author(s): Nicholas Bokulich, Eidgenössische Technische Hochschule Zurich

Review Timeline:

Submission Date:

January 12, 2026

Accepted:

January 24, 2026

Editor: Neha Sachdeva

Reviewer(s): The reviewers have opted to remain anonymous.

Transaction Report:

DOI: <https://doi.org/10.1128/msystems.00023-26>

Re: mSystems00023-26 (HighALPS: Ultra-High-Throughput Marker-Gene Amplicon Library Preparation and Sequencing on the Illumina NextSeq and NovaSeq Platforms)

Dear Dr. Nicholas Andrew Bokulich:

Your manuscript has been accepted, and I am forwarding it to the ASM production staff for publication. Your paper will first be checked to make sure all elements meet the technical requirements. ASM staff will contact you if anything needs to be revised before copyediting and production can begin. Otherwise, you will be notified when your proofs are ready to be viewed.

Cover Image Submissions: If you would like to submit a potential Cover Image, please email a file and a short legend to mSystems@asmusa.org. Please note that we can only consider images that (i) the authors created or own and (ii) have not been previously published. By submitting, you agree that the image can be used under the same terms as the published article. Image File requirements: TIF/EPS, 7.5 inches wide by 8.25 inches tall (at least 2,250 pixels wide by 2,475 pixels tall), minimum 300 dpi resolution (600 dpi preferred), RGB, and no figure elements, e.g., arrows or panel labels. The legend should be a short description of the image, 1-2 sentences recommended. Please download and use this interactive template in Adobe to ensure that your proposed cover image meets our size requirements (<https://journals.asm.org/pb-assets/pdf-text-excel-files/ASM-Interactive-Sizing-Cover-Template-1715689791.pdf>).

Sincerely,
Neha Sachdeva
Editor
mSystems